# The Transcriptomic Response of Cells of the Thermophilic Bacterium *Geobacillus icigianus* to Terahertz Irradiation

**DOI:** 10.3390/ijms252212059

**Published:** 2024-11-09

**Authors:** Sergey Peltek, Svetlana Bannikova, Tamara M. Khlebodarova, Yulia Uvarova, Aleksey M. Mukhin, Gennady Vasiliev, Mikhail Scheglov, Aleksandra Shipova, Asya Vasilieva, Dmitry Oshchepkov, Alla Bryanskaya, Vasily Popik

**Affiliations:** 1Laboratory of Molecular Biotechnologies, Federal Research Center Institute of Cytology and Genetics, Siberian Branch of Russian Academy of Sciences, 10 Akad. Lavrentiev Ave., 630090 Novosibirsk, Russia; sbann@bionet.nsc.ru (S.B.); uvarovaye@bionet.nsc.ru (Y.U.); vasilieva@bionet.nsc.ru (A.V.); alla@bionet.nsc.ru (A.B.); 2Kurchatov Genomics Center of Federal Research Center, Institute of Cytology and Genetics, Siberian Branch of Russian Academy of Sciences, 10 Akad. Lavrentiev Ave., 630090 Novosibirsk, Russia; tamara@bionet.nsc.ru (T.M.K.); mukhin@bionet.nsc.ru (A.M.M.);; 3Budker Institute of Nuclear Physics, Siberian Branch of Russian Academy of Sciences, 11 Akad. Lavrentiev Ave., 630090 Novosibirsk, Russia; m.a.scheglov@inp.nsk.su (M.S.); v.m.popik@inp.nsk.su (V.P.)

**Keywords:** terahertz radiation, *Geobacillus icigianus*, stress response, nonthermal radiation, transcriptomics

## Abstract

As areas of application of terahertz (THz) radiation expand in science and practice, evidence is accumulating that this type of radiation can affect not only biological molecules directly, but also cellular processes as a whole. In this study, the transcriptome in cells of the thermophilic bacterium *Geobacillus icigianus* was analyzed immediately after THz irradiation (0.23 W/cm^2^, 130 μm, 15 min) and at 10 min after its completion. THz irradiation does not affect the activity of heat shock protein genes and diminishes the activity of genes whose products are involved in peptidoglycan recycling, participate in redox reactions, and protect DNA and proteins from damage, including genes of chaperone protein ClpB and of DNA repair protein RadA, as well as genes of catalase and kinase McsB. Gene systems responsible for the homeostasis of transition metals (copper, iron, and zinc) proved to be the most sensitive to THz irradiation; downregulation of these systems increased significantly 10 min after the end of the irradiation. It was also hypothesized that some negative effects of THz radiation on metabolism in *G. icigianus* cells are related to disturbances in activities of gene systems controlled by metal-sensitive transcription factors.

## 1. Introduction

As application areas of nonthermal terahertz (THz) radiation grow [1,2], data accumulate suggesting that this type of radiation can affect not only the rate of cell culture development and biological molecules [3,4,5,6,7], but also cellular metabolic processes overall [8,9,10,11,12].

Moreover, evidence is emerging about changes in intracellular processes under the influence of THz radiation at the transcriptomic level. Many studies show an effect of THz irradiation on the expression of genes in prokaryotic cells. For instance, the use of biological sensors has allowed the determination that 15 min of irradiation of *Escherichia coli* cells by electromagnetic waves of the THz range (power density 0.14 W/cm^2^ and wavelength 130 μm) causes activation of the gene systems of stress defense, disaccharide absorption, and amino acid metabolism that are controlled by transcription factors (TFs) YdeO, ChbR, and TdcR [13,14]. There is also activation of gene networks of oxidative stress and of transition metal homeostasis [15,16], and of biofilm formation processes controlled by a TF called MatA [13].

THz irradiation of *E. coli* K12 JM109 cells for 15 min (power density 0.14 W/cm^2^ and wavelength 130 μm) followed by 10 min of incubation induces genes responsible for the biosynthesis of pili and colanic acid, and genes that control cell division, thereby leading to the formation of cells with abnormal morphology that are prone to aggregation [11].

Mammalian cells also respond to THz irradiation by changing the magnitude of transcription of many genes. For example, exposure of induced human pluripotent stem cells to pulsed radiation of the THz range (~0.8 THz) results in expression changes of genes taking part in the regulation of the actin cytoskeleton and gene networks that are under the control of Zn-dependent TFs, thus leading to cell cycle arrest at the G2–M transition and to disturbances in neuroepithelial cell differentiation pathways [17].

Prolonged pulsed THz irradiation of mouse stem cells causes changes in the expression of genes encoding adiponectin, GLUT4, FABP4, and PPARG; in this context, there is accelerated differentiation of these cells toward the fatty phenotype, and this phenomenon may be explained by activation of the TF gene PPARG (peroxisome proliferator-activated receptor gamma) [18,19,20].

Nonetheless, it is reported [7] that although there has been some progress in the research on the effects of THz radiation on biological systems, there are many technical obstacles on this road, one of which is precise temperature control. It is believed that to detect nonthermal effects of THz radiation, temperature variation should not exceed 0.1 °C [7]. This is a major limitation for the identification of biological effects of THz radiation in temperature-sensitive organisms.

From this standpoint, thermophilic species of bacteria and archaea, which are adapted to high temperatures, are the most suitable for investigation into the effects of THz radiation on living systems because such microorganisms feature high resistance to the effects of temperatures attained during the irradiation.

It has been shown that the thermophilic bacterium *Geobacillus icigianus*, strain G1w1, which has been isolated from sludge samples from a steam/gas hydrothermal (97 °C) vent located near Troinoy Geyser (Valley of Geysers, Kronotsky Nature Reserve, the Kamchatka Peninsula, Russia), is adapted to a wide temperature range from 50 °C to 75 °C with an optimum of ~60 °C [21]. Some features of the metabolism of this strain of *G. icigianus* and its possible industrial use as an oil destructor and producer of 2,3-butanediol have been studied [22,23].

Exposure of *G. icigianus* cells to THz radiation for 15 min (power density 0.23 W/cm^2^, wavelength 130 μm) with strict control of temperature in the medium does not induce heat shock protein genes [12] immediately after the THz irradiation and at 10 min of incubation after its completion. These data indicate the suitability of *G. icigianus* as a model organism for studying the influence of THz radiation at the intracellular level, including the molecular–genetic level.

In the present work, we analyzed transcriptomes of *G. icigianus* cells immediately after THz irradiation (power density 0.23 W/cm^2^ and wavelength 130 μm for 15 min), at 10 min after the end of the irradiation, and in control samples (unirradiated aliquots of the cell culture). A new genome version of *G. icigianus* (GCA_000750005.2) was employed to identify transcripts obtained from RNA libraries. The transcriptomic analysis of the irradiated *G. icigianus* cells showed the absence of activation of heat shock genes, thereby confirming the nonthermal nature of THz radiation. The cells of the thermophilic bacterium *G. icigianus* demonstrated rapid restoration of the activity of gene systems of stress defense of DNA structure and protein structure after the THz irradiation. An exception was the gene systems responsible for transition metal homeostasis that control the export of copper and import of iron; these systems manifested high sensitivity to THz irradiation and the absence of the slightest signs of recovery at 10 min after the exposure cessation.

## 2. Results

A new genome assembly of *G. icigianus* was used, which was constructed by different nucleic-acid sequencing methods to obtain short sequence reads from Illumina NextSeq550 and long reads from MinION. Readers can see (Table 1) that the hybrid assembly of short and long reads (GCA_000750005.2) is less fragmentary than the assembly of only short reads (GCA_000750005.1). For example, the number of contigs longer than 1000 nucleotides in the old assembly was 207, and in the new assembly, it is 13; N50 of the old assembly was 39.9 thousand, and that of the new assembly is 951 thousand (which means that at least 50% of the assembly is covered by contigs at least 951 thousand long). The longest contig in the previous assembly was 181 thousand nucleotides, whereas the longest contig in the new assembly is 1.4 million nucleotides.

The new genome version GCA_000750005.2 was employed to identify transcripts obtained from RNA libraries of (a) samples of irradiated *G. icigianus* cells [0.23 W/cm^2^, wavelength 130 μm, for 15 min], (b) samples of the irradiated *G. icigianus* cells after incubation for 10 min, and (c) samples of control [unirradiated] *G. icigianus* cells.

### 2.1. Alteration of the Transcriptomic Profile of the Thermophilic Bacterium G. icigianus in Response to Irradiation with Electromagnetic Waves in the THz Range (Power Density 0.23 W/cm^2^, Wavelength 130 μm) for 15 min

Cells of the thermophilic bacterium *G. icigianus* responded to the THz irradiation by both increasing (after irradiation log2FoldChange > 1–76, of which 22 are significant; at 10 min after irradiation log2FoldChange > 1–34, of which 14 are significant) and decreasing (after irradiation log2FoldChange < −1–32, of which 30 are significant; at 10 min after irradiation log2FoldChange < −1–158, of which 56 are significant) the expression of genes. Table 2 presents a list of genes whose transcription level was found to be significantly increased after the THz irradiation of the cells for 15 min. One can see that after the irradiation, there is upregulation of the operon-encoding enzymes for the biosynthesis of amino acid histidine (EP10_001935, EP10_001933, and EP10_001940), of the gene responsible for negative regulation of the transport and utilization of N-acetylglucosamine (EP10_000465) [24], the operon encoding the system of transport and utilization of fructose (EP10_002551, EP10_002552, and EP10_002553), a lactic-acid synthesis gene (EP10_003019), and genes of enzymes related to the biosynthesis of bacterial cell wall components (EP10_002881) [25,26,27] and mycothiol (EP10_002879), an analog of glutathione. The latter is involved in the detoxification of alkylating agents and reactive oxygen and nitrogen species and acts as a thiol buffer, which is important for the maintenance of the intracellular environment and for protection against disulfide stress [28,29]. Accordingly, it can be theorized that THz irradiation upsets the stable state of the internal environment in *G. icigianus* cells, and that upregulation of the gene of MshA glycosyltransferase, which initiates an increase in mycothiol synthesis, is aimed at protection from these types of stress.

Similarly, it is possible that the increase in the transcription of the gene encoding phosphomannomutase/phosphoglucomutase (EP10_002881) (involved in the biosynthesis of bacterial cell wall components: lipopolysaccharides [27]) and of the gene coding for the negative regulator of transport and utilization of N-acetylglucosamine (EP10_000465) (promoting inhibition of the utilization of N-acetylglucosamine as a carbon source [24], thereby allowing this compound to be redirected to the synthesis of the main component of the cell wall: peptidoglycan [30]) reflects the negative impact of THz irradiation on the cell wall structure and the need for its restoration. This supposition may be confirmed by the data from proteomic analysis of *G. icigianus*, which indicate overexpression of enzymes for the synthesis of peptidoglycan and its precursor UDP-N-acetylmuramate at 10 min after the end of THz irradiation [12].

Activation of the genes of the PTS system, glycolysis enzyme phosphofructokinase FruB, and a TF called FruR (EP10_002551, EP10_002552, and EP10_002553) (which is a positive regulator of genes of pentose phosphate pathway enzymes and a repressor of glycolysis enzyme genes) indicates, on the one hand, an attempt by the cells to launch the metabolism of sugars, in particular of fructose, and on the other hand, to redirect the flow of carbon through the pentose phosphate pathway. This turn of events (i) may contribute to the production of NADPH via the pentose phosphate pathway, and this compound is necessary for complete protection from oxidative stress [31], whose presence is evidenced by the proteomic analysis of *G. icigianus* cells immediately after THz irradiation, and (ii) can be a response to general inhibition of metabolism in (and growth of) the *G. icigianus* cell under THz irradiation [12].

Furthermore, activation of transcription of the genes responsible for xylose metabolism (EP10_002609), lactic acid synthesis (EP10_003019), and transport of oligosaccharides (EP10_002195) and cyclodextrins (EP10_002265) may be a cell response to THz irradiation in conjunction with the general inhibition of metabolism that is observed during this period at the proteomic level [12] and is aimed at utilizing additional carbon sources [32,33].

Our comparative analysis of the differential activity of genes at 10 min after the end of the irradiation revealed that the number of transcripts of all the proteins presented in Table 2 diminished to the control level and, most likely, their initial upregulation is not implemented at the protein level in the long term. Nonetheless, even brief overexpression of genes of histidine biosynthesis enzymes under the influence of THz radiation during the growth of *G. icigianus* cells on the Luria–Bertani (LB) medium (rich in amino acids) can lead to an excess of this amino acid in the cell, thus suppressing cell growth [34].

The list of genes whose transcription proved to be significantly reduced in *G. icigianus* cells after the irradiation for 15 min is given in Table 3.

From the data in Table 3, it follows that THz radiation negatively influences the transcription of at least five operons. One of them governs copper homeostasis in the cell (EP10_000119, EP10_000120, and EP10_000121) and encodes proteins that ensure the export of copper in case of its excess within the cell. The four others ensure the entry of iron into the cell from different sources and code for various systems responsible for iron import, including siderophore-dependent systems involving proteins YclN-YclO-YclP-YclQ (EP10_000626, EP10_000627, EP10_000628, and EP10_000630), YfmC, YfiYZ (EP10_000812, EP10_000813, and EP10_000814), and YfhA-YusV (EP10_000667 and EP10_000668), as well as transporter proteins of the heme-utilizing system of iron import (EP10_000819, EP10_000820, EP10_000821, and EP10_000822).

The expression of genes encoding a zinc transporter protein (which ensures its import into the cell) (EP10_000839) and a cadmium transporting ATPase (EP10_002301) (which ensures bacterial resistance to toxic effects of cadmium) was also found to be reduced. Therefore, the data presented in Table 3 point to disturbances not only of copper homeostasis, but also of the import of iron and zinc under the action of THz radiation.

The copper ion is an important cofactor for many proteins. It can strongly bind to polar functional groups of amino acid residues in a protein and is capable of switching between oxidation states +1 and +2. When the amount of free copper ions in the cell goes up, this makes the copper ion highly toxic because it undergoes unwanted redox reactions, generating reactive oxygen species, and can bind to proteins inappropriately, thereby altering their properties. The cytotoxicity of the copper ion is reported to be primarily due to its destabilizing impact on iron–sulfur clusters [35,36]. To avoid the potential toxicity of copper ions, organisms have evolved systems intended to strictly control these ions’ concentrations and their intracellular movements to copper target proteins [37,38]. In the Gram-positive soil bacterium *Bacillus subtilis*, excess copper ions in the cytoplasm induce a specific efflux system encoded by the copZA operon [39]. The activity of this system is blocked by THz irradiation in *G. icigianus* cells (Table 3).

As follows from Table 3, at 10 min after the end of the irradiation, the transcriptional activity of operons of almost all copper export and iron and zinc import systems described above remained suppressed in *G. icigianus* cells, and the negative effect of the irradiation not only persisted over time but even intensified. These findings indicated high sensitivity of the copper, iron, and zinc homeostasis systems to THz radiation in *G. icigianus*.

As for genes of GTP 3′,8-cyclase (EP10_003141) and β-barrel assembly-enhancing protease (EP10_003318), we noticed a recovery of the transcription level of these genes at 10 min after THz irradiation cessation; however, even a brief decrease in the biosynthesis performed by these proteins may reflect the processes that occur in the *G. icigianus* cell as a consequence of THz irradiation. For instance, GTP 3′,8-cyclase functions in the chain of biosynthesis of the molybdenum cofactor, which is a part of enzymes that combine proton transfer and electron transfer with the participation of redox centers such as Fe-S, cytochromes, or cofactors FAD/FMN [40,41]. Above, we mentioned the likelihood of disturbances in functions of Fe–S enzymes in *G. icigianus* under the influence of THz radiation as a result of their destabilization under the influence of copper ions. Furthermore, we have previously demonstrated [12] that at the proteomic level, it is components of the electron transfer chain that are most sensitive to THz irradiation. In other words, the diminished transcription of the GTP 3′,8-cyclase gene is just another process associated with the impairment of redox reactions in the bacterial cell after THz irradiation.

Regarding β-barrel assembly-enhancing protease, no data on its functions in Gram-positive microorganisms could be found, but in Gram-negative bacteria, a homolog of this enzyme is the periplasmic zinc metallopeptidase, which takes part in the biogenesis and quality control of the outer membrane [42,43]. The decrease in the biosynthesis of this enzyme immediately after the irradiation may mean problems linked with the possible negative influence of THz radiation on the cell wall structure in *G. icigianus*, as demonstrated at the proteomic level [12].

### 2.2. The Response of the Thermophilic Bacterium G. icigianus Cells at 10 min After the End of THz Irradiation

Table 4 presents a list of differentially expressed genes whose transcription level increased in *G. icigianus* cells 10 min after the end of the irradiation. Readers can see that a delayed reaction of the cells to THz irradiation is upregulation of the operon coding for enzymes of the synthesis of threonine (EP10_002112 and EP10_002113), whose excess, just as an excess of histidine (see Table 2) [34], inhibits cell growth [44]. A similar toxic effect on the growth of B. subtilis cells has been registered during an excess of such amino acids as glutamate and serine [45,46,47].

We have shown earlier that at the proteomic level, irradiation of *G. icigianus* cells diminishes the level of L-serine deaminase, which helps to utilize serine, and upregulates serine hydroxymethyltransferase, which synthesizes serine [12]; these phenomena undoubtedly contribute to the accumulation of serine in the cell.

Taken together, the data from transcriptomic and proteomic assays of *G. icigianus* cells indicate that THz irradiation promotes excessive accumulation of such amino acids as histidine, serine, and threonine, which, as shown before [34,44,45,46,47], have a toxic effect on cell growth; the underlying mechanisms may involve disturbances of the cell wall structure. In particular, an excess of serine in the cell increases the probability of its incorporation—instead of alanine—into peptidoglycan, thereby impairing peptidoglycan synthesis and weakening the cell wall [48].

It also follows from Table 4 that 10 min after the end of the irradiation, there is greater transcription of the carboxylic acid transporter gene (EP10_002156) and genes encoding transporters of glutamine and other amino acids (EP10_000138, EP10_001536, EP10_002622, EP10_00263, EP10_002624, and EP10_002625). These data indicate the beginning of a recovery of systems responsible for the utilization of amino acids and of other carbon sources from the environment; this is because at the protein level, at this time point, there is underexpression of proteins that ensure the transport of amino acids into the cell, which are used by the cell as energy sources during growth in the LB medium [12].

Another consequence of THz irradiation of *G. icigianus* cells is enhanced transcription of genes coding for pyrrolidone-carboxylate peptidase (EP10_002477) and 5-oxoprolinase (EP10_000693). Pyrrolidone carboxyl peptidase, also known as pyroglutamyl peptidase, is an enzyme that plays a key role in the hydrolytic removal of noncanonical amino acid L-pyroglutamate from N termini of peptides and proteins, thus yielding free L-pyroglutamate, also known as 5-oxoproline [49,50,51]. L-Pyroglutamate forms via spontaneous cyclization of glutamine and glutamate [52], including when their residues are located at N termini of polypeptides and glutaminylpeptide cyclotransferase is involved [53]. That is, L-pyroglutamate is a product of damage and spontaneously arises always and everywhere; its accumulation negatively affects cell growth [52]. Free L-pyroglutamate (5-oxoproline) is metabolized by 5-oxoprolinase and can be used by some bacteria, including B. subtilis, as a nitrogen source [54]. Thus, the overexpression of genes encoding pyrrolidone-carboxylate peptidase and 5-oxoprolinase may indicate an increase in metabolite damage to proteins and peptides under the action of THz irradiation.

As for transcriptional regulator BetI, its homologs in various bacterial species are sensors of osmotic stress and activate the transcription of genes of the small regulatory RNAs that control the activity of the quorum-sensing system, which helps to regulate the response to osmotic stress [55,56,57]. It is possible that in *G. icigianus*, too, the reason for the stronger transcription of BetI under THz irradiation is osmotic stress resulting from impairment of the exchange of metabolites, including amino acids and carboxylic acids, between the cell and its environment, as discussed above.

Table 5 shows a list of differentially expressed genes whose transcription level diminished in *G. icigianus* cells 10 min after the end of the irradiation. One can see that a consequence of the exposure of *G. icigianus* cells to THz radiation is a decline in the activity of at least three operons, one of which encodes enzymes for the catabolism of branched-chain amino acids (EP10_001408, EP10_001409, EP10_001411, and EP10_001412) [58], which is a system participating in maintenance of the energetic and metabolic state of the *G. icigianus* cell; in the irradiated cell, according to our previous proteomic analysis, this state is depressed at this time point [12]. Consequently, the time that elapsed since the end of the irradiation (10 min) was not enough to restore the activity of this system.

Ten minutes after the end of THz irradiation, in *G. icigianus* cells, there was underexpression of the operon that encodes proteins of the two-component system of protein phosphorylation on arginine [arginine kinase mcsB (EP10_003551) and arginine kinase activator protein mcsA (EP10_003550)], repressor CtsR (EP10_003549), negative regulator of genetic competence ClpC/MecB (EP10_003552), and the RadA protein (EP10_003553), which participates in DNA repair.

Arginine is a key amino acid in protein–protein and DNA–protein interactions, and phosphorylation of arginine residues by McsB is critical for tagging and subsequent degradation of proteins [59], for modulation of the cellular response to oxidative stress [60,61], and for such phenomena as cell motility and competence [62]. The main targets of McsB-mediated arginine phosphorylation are heat shock repressors CtsR and HrcA and major components of the protein quality control system, e.g., protease ClpCP and chaperone GroEL [63]. McsB is triggered by autophosphorylation of certain arginine residues, especially during heat stress, thus allowing for controlled degradation of repressor CtsR by protease ClpCP [64]. A reduced activity of this operon and of the Clp protease gene (EP10_001951) indicates some repression of the stress defense systems protecting DNA and proteins from damage. A similar response to THz irradiation was documented here for genes of chaperone ClpB (EP10_000019), catalase KatG (a marker of oxidative stress) (EP10_002431), and universal stress protein (EP10_001011).

Finally, 10 min after the end of THz irradiation, there was diminished transcription of the operon whose genes—divIC (EP10_001436), yabQ (EP10_001437), and yabP (EP10_001438)—encode proteins taking part in ectospore formation. For instance, the DivIC protein is required for the formation of septa under conditions of vegetative growth and spore formation, as well as for the induction of genes expressed under the control of sporulation TFs σF and σE [65,66], whereas proteins YabQ and YabP directly participate in the formation of the spore coat [67,68,69].

The decreased transcription of the gene of the Btr factor (EP10_003512)—which positively controls the activity of the iron absorption system with the participation of a siderophore called bacillibactin [70]—points to the existence of another component of THz irradiation’s negative impact on the homeostasis of transition metals.

Regarding the reduction in the transcription of the gene (EP10_00008) of anhydro-N-acetylmuramic acid kinase (an enzyme of the peptidoglycan recycling pathway in various bacteria) [71,72], this is another aspect of the negative influence of THz irradiation on the cell wall structure in *G. icigianus*, as shown at the proteomic level [12].

### 2.3. Bioinformatic Analysis of Regulatory Regions of Operons in the Genome of G. icigianus That Are Most Sensitive to THz Radiation

It was stated above that the most sensitive to THz irradiation were at least five operons involved in the homeostasis of transition metals (copper, iron, and zinc); the activity of these operons decreased immediately after the irradiation, and this negative effect not only persisted over time but even strengthened. These were found to be operons that control the export of copper from the cell and the import of iron and zinc into the cell (Table 3 and Table 6).

These findings suggested that THz irradiation impairs the expression of these operons through activation of metal-dependent transcriptional repressors that control these systems. We performed a bioinformatic analysis of regulatory sequences of the operons that were most sensitive to THz irradiation (see Table 3 and Table 6) by the phylogenetic footprinting method [73], which allows the identification of the regions in promoters of operons that are conserved among different species of microorganisms and are associated with specific TFs. In this case, the analysis was carried out among bacteria belonging to the phylum Bacillota (synonym: Firmicutes). As a result, in intergenic sequences of *G. icigianus* operons whose protein products implement the import of iron and zinc, regions were identified that are conserved among Firmicutes taxa and contain motifs of potential binding sites for TFs from the Fur family (Fur, Zur, and PerR), whereas in the regulatory sequence of the operon participating in the control of copper export, motifs were found from binding sites for TFs CsoR and Fur/PerR.

To clarify the location of the binding sites for TFs in intergenic regions of operons, we also simulated recognition of promoters by σ70 with the help of the Promotech V1.0 software (https://github.com/BioinformaticsLabAtMUN/PromoTech, accessed on 21 June 2024) and identified potential binding sites of the σ70 subunit of RNA polymerase (−35/−10) using BPROM [74] (http://www.softberry.com/berry.phtml?topic=bprom&group=programs&subgroup=gfindb, accessed on 21 June 2024) [75]. The localization of binding-site motifs for TFs, of positions −35/−10, and of promoters specific for σ70, was determined in accordance with genome landmarks (+1 from the beginning of a CDS). The data are presented in Table 6.

As one can see in the data in Table 6, in promoters of operons 319, 438, 442, and 454, which control the import of iron and zinc in *G. icigianus*, there are multiple motifs of potential binding sites belonging to the Fur family of metal-dependent TFs, which are located either within the sequence of a binding site of the σ70 subunit of RNA polymerase or at some distance. The fact that they were found in regions conserved among other Firmicutes taxa suggests these sites are likely functional. As for such sites in the operator region of promoters of operons 438, 442, and 454, their location is consistent with the classic role of these TFs as transcription repressors in the presence of divalent metal ions. Nonetheless, no such sites were identified in the operator region (−53/−13) of the promoter of operon 319. This means that *G. icigianus* may still have them in a modified form, but the presence of Fur-like sites in a fairly distant promoter region in all these operons possibly indicates the existence of additional mechanisms underlying the regulation of these operons in *G. icigianus*. It is known that in other bacteria, Fur can serve as a repressor and as an activator through sites quite distant from the operator [76], e.g., as a repressor in the absence of metal ions [77]. That is, it is possible that in *G. icigianus*, which lives under conditions of mineralized hot springs [78], various mechanisms have arisen for fine-tuning the expression of Fur-dependent operons during an excess or deficiency of metal ions.

We can hypothesize that under the action of THz irradiation, repression of the above-mentioned operons that control the import of iron and zinc is mediated by TFs of the Fur family in two stages and through different mechanisms. At the first stage, THz irradiation stimulates the transition of Fe^3+^ to Fe^2+^ and of other metals—which interact with TFs of this family—into the divalent state, thereby enhancing the binding of a TF to DNA, and consequently the repression of the operon through sites located in the operator zone of the promoter (Figure 1). The resulting decrease in intracellular levels of iron and zinc causes even stronger inhibition of Fur-dependent transcription through sites that act as repressors during an iron and/or zinc deficit.

The finding that the irradiation significantly affects the expression of only these Fur/PerR/Zur-dependent operons may be due precisely to the existence of different mechanisms of regulation of Fur-dependent operons in *G. icigianus* and to characteristics of their binding sites. These sites, though similar in structure, may have different affinity levels for certain TFs, whose influence on transcription may depend, among other things, on external factors. For example, oxidative conditions affect the activity of PerR as a transcriptional repressor. In *B. subtilis*, oxidative dominance in the cell drives inactivation of PerR and its subsequent degradation [79,80], which creates conditions for derepression of Fe^2+^–PerR-dependent operons. Above, we mentioned the possible transition of Fe^3+^ to Fe^2+^ under the influence of THz irradiation, and this transition can also serve as a catalyst for Fenton reactions leading to the reduction of hydrogen peroxide and the emergence of highly reactive hydroxyl radicals provoking oxidative stress [81]. The likelihood of oxidative conditions in the *G. icigianus* cell under the influence of THz irradiation is also evidenced by our results on downregulation of the gene of catalase KatG (oxidative stress sensor) at 10 min after the end of the irradiation (Table 5). In other words, THz irradiation creates conditions for derepression of Fe^2+^-dependent operons regulated through sites that have stronger affinity for TF PerR than for other members of this family.

Furthermore, it is possible that the differences in the expression of Fur/PerR/Zur-dependent operons may be due to the presence of other, unknown regulators in *G. icigianus*. Indeed, in promoters of the aforementioned Fur-dependent operons of *G. icigianus*, conserved regions were also identified that are not associated with motifs of binding sites of the TFs available for our analysis.

Regarding operon 75 (*csoR*-*copAZ*), which controls copper export, many conserved regions were identified in its regulatory region, one of which is associated with a binding site of copper-dependent TF CsoR. The localization of this site within the binding site of the σ70 subunit of RNA polymerase is characteristic of this TF as a transcription repressor in the absence of copper ions. Other conserved regions are associated with binding-site motifs of Fe^2+^-dependent TFs from the Fur family and are located quite far from a potential σ70-dependent promoter (Table 6).

The presence of a binding site for TF CsoR implies its participation in the regulation of the expression of both the operon and its own gene, as reported for this operon in other *Bacillus* species [37]. In the absence of copper ions, CsoR is bound to the promoter and blocks the expression of the operon, whereas the interaction of CsoR with copper ions weakens its affinity for promoter DNA and causes activation of the operon. That is, if we assume that as a result of THz irradiation, the export of copper ions is blocked via a decrease in the number of transcripts of exportase CopA (and there is accumulation of copper ions in the cell), then an even greater decrease in the number of all transcripts encoded by this operon after the irradiation cessation does not appear to be logical. This is because the accumulation of Cu should drive transcriptional activation of the operon. The detection of multiple motifs of binding sites for TFs of the Fur/PerR family in the regulatory region of the *csoR*-*copAZ* operon in *G. icigianus* helps to clarify the possible reason for such strong repression of the *csoR*-*copAZ* operon after THz irradiation during a deficit of iron ions.

## 3. Discussion

Examination of the data presented in Table 2, Table 3, Table 4 and Table 5 suggests that at the transcriptome level, the specific features of the response of the thermophilic bacterium *G. icigianus*’s cells to THz irradiation are primarily attributable to disturbances in the expression of genes of the copper, iron, and zinc homeostatic systems, namely, to blockage of the transcription of genes coding for the systems of copper export and iron and zinc import. Additionally, 10 min after the end of the irradiation, the activity of these systems did not recover in *G. icigianus* cells; moreover, their inhibition got stronger. The high sensitivity of systems of transition metal homeostasis to THz radiation is shown by *E. coli* cells too [16]. These findings indicate that dysfunction of proteins having Fe/S clusters may emerge in the cell owing to the accumulation of excess copper ions and lack of iron. Indeed, studies of the effects of general copper toxicity have shown that the predominant intracellular form of copper—Cu(I)—destabilizes iron–sulfur cofactors that are weakly bound to dehydratases. Cadmium ions have a similar impact on dehydratases [82]; the expression of genes encoding the system of cadmium export from the cell is also low in *G. icigianus* after THz irradiation (Table 3). Dihydroxy acid dehydratase (IlvD) and isopropyl malate dehydratase (LeuC) in the branched-chain amino acid synthesis pathway can be affected by this phenomenon, as can fumarase A (FumA) and 6-phosphogluconate dehydratase (Edd) [33]. 6-Phosphogluconate dehydratase is a key enzyme of the pentose phosphate cycle, producing cofactor NADPH, which is necessary to ensure biosynthesis processes in the cell that involve NADPH-dependent enzymes, including the synthesis of leucine, valine, isoleucine, lysine, and other amino acids. It is likely that the increase in the protein level of this enzyme in *G. icigianus* cells 10 min after the irradiation [12] may be connected to, among other things, a decrease in the enzymatic activity owing to destabilization of the active center.

Another important contributor to the adverse impact of THz radiation on *G. icigianus* cells is probable emergence of a toxic effect of some amino acids’ excess on cell growth, as shown for histidine, threonine, and serine [34,44,45,46,47]. At the transcriptomic level, after the exposure of *G. icigianus* cells to THz radiation, we see overexpression of operons encoding enzymes for the synthesis of histidine and threonine (Table 2 and Table 4), and at the proteomic level, overexpression of enzymes synthesizing serine and downregulation of enzymes that help to utilize it [12].

During cell cultivation in a medium rich in amino acids, the observed effects can undoubtedly lead to the accumulation of these amino acids in the cell, and the underlying mechanisms may be related, among other things, to disturbances of the structure of the cell wall. For instance, for serine, it has been demonstrated that its negative effect can be ascribed to possible replacement of alanine by serine during the synthesis of peptidoglycan, resulting in a disturbance of the cell wall structure [48]. A confirmation of the presence of these aberrations is the cell response involving an increase in the biosynthesis of cell wall components. At the transcriptomic level, an example is changes in the expression of EP10_002881 and EP10_000465 genes (Table 2), the products of which take part in the biosynthesis of lipopolysaccharides and peptidoglycan [24,27,30]. At the proteomic level, an example is upregulation of enzymes synthesizing peptidoglycan and its precursor UDP-N-acetylmuramate [12]. It is probable that in *E. coli*, too, some of the cell wall disturbances observed after THz irradiation [11] are partly a consequence of activation of the gene systems that govern serine transport and metabolism through TF TdcR [14].

The adverse effect of THz radiation on cell wall structure and *G. icigianus* growth is also supported by data on a decrease in the transcription of genes of anhydro-N-acetylmuramic acid kinase (EP10_00008) and of β-barrel assembly-enhancing protease (EP10_003318) involved in peptidoglycan recycling [71,72] and in the biogenesis and quality control of the outer membrane [42,43], respectively, as well as downregulation of the operon whose genes (EP10_001436, EP10_001437, and EP10_001438) code for proteins that modulate vegetative growth and sporulation [65,66].

Concerning genes of the operon related to control over levels of damage to proteins and DNA (EP10_003549, EP10_003550, EP10_003551, EP10_003552, and EP10_003553), the reduction in their activity and the activity of the protease Clp gene (EP10_001951) in *G. icigianus* indicates some inhibition of systems responsible for stress defense of DNA and proteins (Table 5). This finding is also evidenced by weaker transcription of several stress protein genes, including genes of chaperone protein ClpB (EP10_000019), catalase KatG (a marker of oxidative stress) (EP10_002431), and universal stress protein (EP10_001011) (see Table 5). On the other hand, 10 min after THz irradiation, we observed recovery of mRNA expression of MshA glycosyltransferase (EP10_002879) (which initiates an increase in the synthesis of mycothiol (an analog of glutathione), which participates in the detoxification of alkylating agents and reactive oxygen species and in protection from disulfide stress) [28,29]. Furthermore, in all the irradiated groups, there was no activation of *G. icigianus* systems of genes responsible for protection from DNA damage (for example, EP10_000368: RecG, EP10_000478: RecA, EP10_001399: RecN, EP10_001304: RecO, EP10_001488: RecF, and EP10_001472: RecR) or participating in the SOS response (EP10_000509: LexA and EP10_000685: SRAP, SOS response-associated peptidase), as well as in other systems of the oxidative-stress response (EP10_000843 and EP10_002144). The above results imply certain resistance of *G. icigianus* cells to this type of radiation.

At the proteomic level, cells of the thermophilic bacterium *G. icigianus* also turned out to be quite resistant to stress and demonstrated a rapid recovery (to the control level) of systems possessing chaperone, protease, nuclease, and antioxidant activities [12]. On the contrary, in mesophilic *E. coli*, gene systems of stress defense remained active at 10 min after THz irradiation [13,15].

Thus, these data confirm the nonthermal nature of THz radiation and point to some inhibition of antistress systems protecting DNA and proteins from damage in *G. icigianus* cells after irradiation. However, after THz irradiation, there was a rapid recovery of the activity of the gene of MshA glycosyltransferase, which participates in the detoxification of alkylating agents and reactive oxygen species and in protection from disulfide stress; furthermore, there was no activation of a number of gene systems for protecting cells from DNA damage (RecG–RecA–RecN–RecO–RecF–RecR–LexA–SRAP). These findings indicate sufficient stress resistance of *G. icigianus* to THz radiation.

The obtained data also demonstrated that THz irradiation can contribute to the emergence of the toxic stress related to overexpression of genes coding for enzymes of threonine and histidine biosynthesis, the excess of which suppresses cell growth. A THz radiation-induced decrease in the expression of genes involved in peptidoglycan recycling, and in biogenesis and quality control of the outer membrane, may also negatively affect the growth and division of *G. icigianus* cells.

It was found that gene systems of the transport of copper, iron, zinc, and cadmium are extremely sensitive to THz radiation, and that blockage of their expression by THz radiation may be a harbinger of the toxic stress that leads to destabilization of the function of proteins, including those containing Fe/S clusters. It can be theorized that some negative effects of THz irradiation on metabolism in cells of the thermophilic bacterium *G. icigianus* can be explained by disturbances in the activity of gene systems controlled by metal-sensitive TFs.

## 4. Materials and Methods

### 4.1. Cultivation of the Cells

To conduct the experiments, strain G1w1 of the Gram-positive thermophilic bacterium *G. icigianus* was used, from the collection at the ICG SB RAS. The bacterial culture was grown in the LB medium of the following composition: 5 g of NaCl, 5 g of yeast extract, and 10 g of tryptone per 1000 mL of the medium. To conduct a series of reproducible experiments, we employed a previously described method for obtaining cell culture in the logarithmic growth phase [12]. A bacterial culture that was grown from a single colony and reached an optical density of 0.5 was frozen in 400 μL portions in 10% glycerol for storage at −70 °C and subsequent use in experiments. From one aliquot of the frozen culture, 200 μL was added to each 5 mL portion of the LB medium. After 18 h of cultivation, the cells were reseeded into a culture flask containing 100 mL of the LB medium. Cultivation was carried out at 60 °C and 250 rpm in an orbital incubator. For the transcriptomic analysis, the culture was grown to an optical density of 0.5, corresponding to the mid-logarithmic growth phase. To collect material for one biological replicate, the *G. icigianus* culture was grown in three flasks at 3 h intervals.

### 4.2. Irradiation of Cultured Cells

The experiments were carried out by means of a THz free electron laser (Novosibirsk Free Electron Laser facility) at the Siberian Synchrotron and Terahertz Radiation Centre, which was designed and launched by Budker Institute of Nuclear Physics SB RAS (Table 7).

The experimental scheme is shown in Figure 2.

For the experiment, 60 μL of the bacterial culture was transferred to a specially designed cuvette, which enables simultaneous irradiation of the entire volume of the sample [16]. Due to the small amount of the material that could be collected after one irradiation session, each experimental sample was prepared as a pool of samples. One biological replicate was defined as material collected from the three consecutive culture flasks at 3 h per flask during 1 day of irradiation. The cells were exposed to THz radiation as described by [16] at power density 0.23 W/cm^2^ and wavelength 130 μm, while the temperature of the medium in the cuvette was controlled with the help of a TKVr-SVIT101 thermal imager and maintained at 60 °C ± 2 °C. The duration of irradiation of each aliquot was 15 min. The culture was then collected from the chamber and either frozen in liquid nitrogen to study the rapid response or transferred to a 60 °C thermostat and kept there for 10 min to develop the response, after which the cells were frozen in liquid nitrogen. As a control, 60 μL of cells was transferred to a specially designed cuvette, which was placed in a thermostat at 60 °C and incubated for 15 min. Then, the culture was collected from the cuvette and either frozen in liquid nitrogen or transferred to a thermostat at 60 °C and incubated for 10 min, after which the cells were frozen in liquid nitrogen.

### 4.3. Whole-Genome Sequencing

To isolate high-molecular-weight DNA, *G. icigianus* G1w1 was inoculated into a liquid LB medium; cultivation time was 24 h. The biomass was collected into separate tubes, lysed by means of 10% SDS in TE buffer, and gently mixed without shaking. A phenol–chloroform mixture (1:3) was added and gently mixed, and the suspension was centrifuged for 10 min at 16,100× *g*. The aqueous fraction was transferred into a clean test tube, 1 mL of isopropyl alcohol was introduced, and the test tube was shaken. DNA was collected with a sterile wooden stick, washed in 96% ethyl alcohol and then in 70% ethyl alcohol, air-dried, and dissolved again in TE buffer. The A_260_/A_280_ ratio of the sample was assessed by a spectrophotometric method on an Epoch instrument (BioTek Instruments, Winooski, VT, USA); the ratio was 1.95. DNA concentration was determined fluorometrically on a Qubit fluorimeter (the DNA BR kit, Qubit, Thermo Fisher Scientific, Waltham, MA, USA); the concentration was found to be 854 ng/μL. Next, sample preparation was carried out with the Rapid Sequencing kit (Oxford Nanopore Technologies, Oxford, UK) in accordance with the manufacturer’s instructions, and the DNA was sequenced on a MinION device (Oxford Nanopore Technologies, UK).

To extract DNA for whole-genome sequencing on an Illumina instrument, a DNA extraction kit from Qiagen (DNeasy Blood & Tissue Kits, Qiagen, Germantown, MD, USA) was used according to the manufacturer’s instructions. Six biological replicates were set up for the sequencing. The A_260_/A_280_ ratio in the samples was assessed by the spectrophotometric method on an Epoch instrument (BioTek Instruments, USA); the ratios were in the range of 1.8–2.0. DNA concentration was determined fluorometrically on a Qubit fluorimeter (the DNA BR Kit, Qubit, USA); concentrations were >50 ng/μL. DNA fragmentation was performed by ultrasonication on a device from Covaris, LLC (Woburn, MA, USA). The DNA was then sequenced on an Illumina NextSeq550 instrument.

### 4.4. Genome Assembly and Annotation

To assemble the genome, short reads from Illumina NextSeq550 and long reads from MinION were used. The hybrid assembly was performed in Unicycler v0.4.8 software [83]. The assembly was annotated in Prokka 1.14.6 [84]. Using QUAST v5.0.2 software [85], the new assembly was compared with the previous assembly (GCA_000750005.1).

### 4.5. RNA Sequencing

The impact of THz radiation on the cell transcriptome was analyzed by RNA sequencing on the Illumina massively parallel sequencing platform. Total RNA from bacterial cells was isolated by means of the Qiagen RNA Mini Kit, with modifications to the manufacturer’s protocol. Namely, to a frozen cell pellet, 100 μL of the RNAprotect Bacteria Reagent was added. After thawing, the cells were carefully resuspended by pipetting and incubated for 5 min at room temperature. The cells were pelleted by centrifugation for 3 min at 4000× *g*. Next, 80 μL of TE buffer containing 15 mg/mL lysozyme was added to the precipitate, which was resuspended and incubated for 5 min. After that, 10 μL of a proteinase K solution was added to the cells with incubation for 5 min. Next, 5 μL of 10% SDS was added to the cells, and the cells were ground in an Eppendorf tube by means of a plastic pestle. After that, lysis was carried out by the addition of 700 μL of Buffer RLT with mercaptoethanol followed by pipetting, after which 500 μL of 80% EtOH was introduced, and after mixing, all the liquid was applied to the column and centrifuged for 30 s at 8000× *g*. The column was washed with 700 μL of Buffer RW1, after which the column was transferred to a new Collection Tube and washed twice with 500 μL of Buffer RPE. Then, RNA was eluted with 30 μL of RNase-free water. After the isolation, the quality of the RNA was assessed on a BA2100 bioanalyzer using the RNA Nano kit. The isolated RNA showed no signs of degradation (Figure 3), and RNA concentration was determined fluorometrically on the Qubit instrument with the RNA High Sensitivity Kit.

To study changes in gene transcription under the influence of THz radiation, RNA-Seq libraries of mRNA were created next. To remove ribosomal RNA (by the Ribo-Depletion procedure), the RiboMinus™ Transcriptome Isolation Kit bacteria (Invitrogen, Thermo Fisher Scientific, Waltham, MA, USA) were used in strict accordance with the manufacturer’s protocol. Magnetic beads were dispensed at 250 μL of the initial suspension per 1.5 mL Eppendorf tube and washed twice with 250 μL of H_2_O on a magnetic rack and once with 250 μL of Hybridization Buffer. Then, they were dispersed in 100 μL of Hybridization Buffer. To carry out the reaction, 1.5 μg of the total bacterial RNA isolated above was used in a 20 μL solution, to which 4 μL of RiboMinus™ Probe and 100 μL of Hybridization Buffer were added. The mixture was incubated on a thermal cycler for 5 min at 37 °C, then on ice for 30 s, and added to the suspension of magnetic beads with immediate stirring. Next, incubation was performed for 15 min on a thermoshaker at 37 °C, after which the beads were immediately placed on a magnetic stand, and the liquid fraction containing ribo-depleted RNA was collected; final purification and concentration were carried out on AMPure RNA Clean magnetic beads. The absence of considerable RNA degradation and the presence of impurities of ribosomal RNA less than 20% (in the range of 12.4–17.2%) were monitored on the BA2100 bioanalyzer with the RNA Pico Kit.

To generate 12 barcoded RNA-Seq libraries, the TruSeq Stranded mRNA Library Prep Kit was utilized according to the manufacturer’s protocol with several modifications. Namely, 10–15 ng of ribo-depleted RNA was used directly from the RNA fragmentation stage; the preceding selection steps were omitted. RNA in a 4.3 μL solution was added to 4.3 μL of the Fragment, Prime, Finish Mix; fragmentation was conducted for 8 min, and subsequent steps were performed in accordance with the manufacturer’s protocol. Twelve PCR cycles were used to amplify the libraries; final purification was carried out on AMPure XP magnetic beads. TruSeq RNA Single Indexes Set B was chosen. The quality of the resulting libraries and their molarity were checked on the BA2100 Bioanalyzer with the DNA High Sensitivity Kit; the libraries were diluted 1:10 before application to the bioanalyzer (Figure 4). The molarity of the obtained libraries was in the range of 20–45 nM.

Sequencing of the resultant libraries was performed on a NextSeq550 device, with a read of 75 bp, and the average amount of data per library was 38 million sequence reads.

To identify genes differentially expressed in *G. icigianus* cells (Firmicutes) as a consequence of THz irradiation, the obtained Illumina nucleotide reads were preprocessed and mapped. At the preprocessing stage, quality was evaluated in FastQC software v.0.11.9 (http://www.bioinformatics.babraham.ac.uk/projects/fastqc, accessed on 30 November 2019)), and filtering was conducted using TrimGalore 0.6.7 (https://www.bioinformatics.babraham.ac.uk/projects/trim_galore/, accessed on 6 May 2021). The cleaned reads were then mapped to the new hybrid version of the *G. icigianus* reference genome assembly. The mapping was performed by means of Star software v.2.7.10a [86]. As a result, files of the BAM format (a binary equivalent of the SAM, Sequence Alignment/Map format) were obtained. Relative activity of each gene was calculated according to that gene’s nucleotide read coverage on the reference genome after mapping of each library.

Coverage was determined with the htseq-count tool from HTSeq software v.0.6.1 [87], using a gtf file with full-annotation gene coordinates and an indexed bam file obtained via the mapping of nucleotide reads. The mapping data for each library were next compiled into a single table, which was subjected to further analysis. Statistical analysis of differential expression was performed in DeSeq2 v.1.16.1 [88]. Each library represented 30 to 40 million reads, with more than 99% of the reads mapped to the genome.

Using the obtained data on relative gene expression, by means of tools from the DeSeq2 v1.30.1 software, principal component analysis was performed, and the distances between samples were visualized in the respective graph. The results showed that THz-irradiated samples clustered separately from controls, both in the group of samples with 10 min incubation and in the group without the incubation, suggesting that THz radiation was the main reason for the difference in gene expression profiles between the groups.

### 4.6. Phylogenetic Footprinting and Recognition of Promoters Specific to σ70

For bioinformatics analysis of the regulatory sequences in the above-mentioned operons, we employed the MP3 v1.01 software [73], which is based on phylogenetic footprinting [71]. As a promoter, we regarded a region of genomic sequence −500…+1 relative to an annotated CDS of the initial gene of an operon; if a neighboring upstream operon was closer than 500 bp, then we used the sequence region between the annotated CDS of the initial gene of the analyzed operon and the CDS of the last gene in the neighboring upstream operon. According to a described procedure [73], lists of orthologous genes were prepared with an orthology detection tool called GOST [89] in all available reference genomes from the same phylum (Bacillota) to which the genome under study belongs. For subsequent identification of promoter regions in each orthologous gene, data on operon structure of known bacterial genomes from the DOOR2.0 database [90] were used, as were corresponding genomes with annotations from the NCBI database [91]. This approach allowed compilation of the most complete sample of promoter regions of orthologous genes for subsequent MP3 calculations. We modified the above procedure [73] of selection of operon promoters into the input sample to achieve robustness of the obtained results: we made the requirements more stringent by selecting only those genomes in which all orthologous genes contained in a target operon are present. In the event that a gene contained in a target operon did not share sufficient similarity with any known bacterial proteins, it was excluded from the analysis. This approach made it possible to substantially improve the reproducibility of the calculation results. The identified motifs were compared with sets of known bacterial TF-binding sites from the PRODORIC database [92]; the Tomtom tool v5.5.5 from the MEME v5.5.5 software suite [93] was utilized for the comparison; similarity was considered significant at an E-value of <0.05.

To identify binding sites for σ70, we used two recognition software packages: Promotech V1.0 (a machine-learning-based method for promoter recognition in a wide range of bacterial species) and BPROM [74] (bacterial σ70 promoter recognition software with ~80% accuracy and specificity). The latter enables detection of the −10 box and −35 box in recognized promoters. All programs were launched with default settings; the obtained match between recognition results of these two programs allowed us to achieve high final accuracy of recognition.

## Figures and Tables

**Figure 1 ijms-25-12059-f001:**
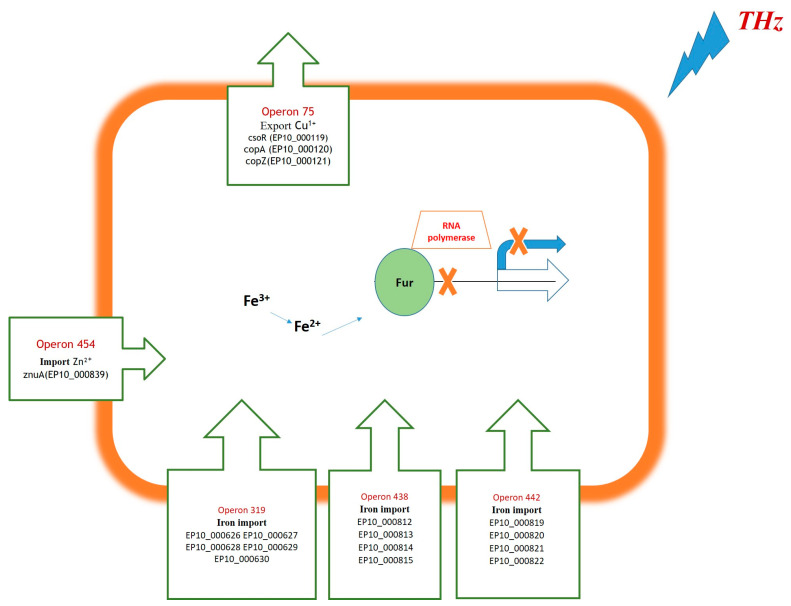
Proposed mechanism of action of THz irradiation on Fur-dependent operons in *G. icigianus*.

**Figure 2 ijms-25-12059-f002:**
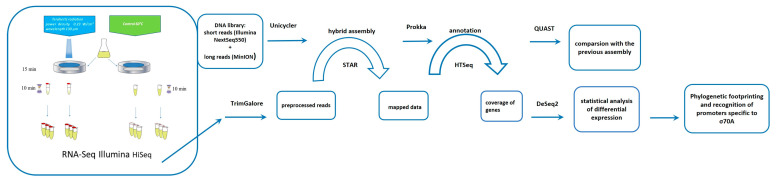
The experimental scheme.

**Figure 3 ijms-25-12059-f003:**
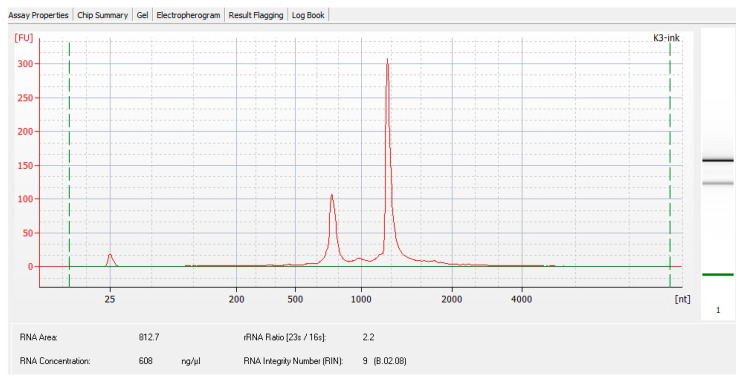
RNA quality of total RNA from the K3-ink sample.

**Figure 4 ijms-25-12059-f004:**
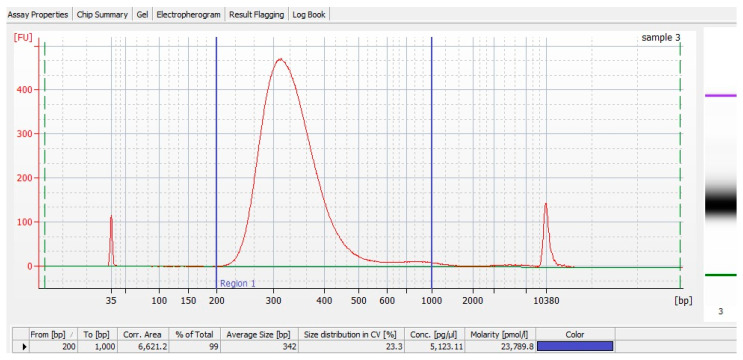
An example of the quality of an obtained RNA-Seq library.

**Table 1 ijms-25-12059-t001:** Metrics determined using the QUAST software (version 5.0.2).

Assembly	GCA_000750005.1	GCA_000750005.2
Number of contigs (≥0 bp)	207	13
Number of contigs (≥1000 bp)	207	13
Number of contigs (≥5000 bp)	112	8
Number of contigs (≥10,000 bp)	84	8
Number of contigs (≥25,000 bp)	48	8
Number of contigs (≥50,000 bp)	17	7
Total length (≥0 bp)	3,457,810	3,649,981
Total length (≥1000 bp)	3,457,810	3,649,981
Total length (≥5000 bp)	3,242,398	3,643,168
Total length (≥10,000 bp)	3,042,086	3,643,168
Total length (≥25,000 bp)	2,474,700	3,643,168
Total length (≥50,000 bp)	1,402,509	3,616,816
# contigs	207	13
Largest contig	181,745	1,416,833
Total length	3,457,810	3,649,981
GC (%)	52.03	51.47
N50	39,998	951,425
N75	20,230	397,370
L50	25	2
L75	54	3
Number of N’s per 100 kbp	0	0

**Table 2 ijms-25-12059-t002:** Genes whose transcription levels significantly rise (padj < 0.1) immediately after THz irradiation of *G. icigianus* cells for 15 min.

Locus ID	log2Fold Change	padj	Protein/Enzyme
EP10_000465	0.98031	0.028035	HTH-type transcriptional repressor NagR: involved in transport and utilization of N-acetylglucosamine
EP10_001935	1.295636	0.007308	ATP phosphoribosyltransferase regulatory subunit/step 1/9 histidine biosynthesis
EP10_001937	1.060575	0.083198	Histidinol dehydrogenase/L-histidinol + H_2_O + 2NAD^+^ = 3H^+^ + L-histidine + 2NADH; Cofactor Zn^2+^, step 9/9
EP10_001940	0.923879	0.063759	Imidazole glycerol phosphate synthase subunit HisH, step 5/9, histidine biosynthesis
EP10_002195	1.213861	0.014874	Oligosaccharide import ATP-binding protein MsmX
EP10_002265	1.492904	0.046696	Cyclodextrin-binding protein
EP10_002551	3.56812	2.92 × 10^−16^	PTS system fructose-specific EIIABC component, EC 2.7.1.69
EP10_002552	3.419195	4.25 × 10^−12^	Phosphofructokinase-1, EC 2.7.1.56 Fru-6-P + ATP = fructose-1,6-bisphosphate + ADP
EP10_002553	3.419688	3.18 × 10^−14^	HTH-type transcriptional repressor FruR
EP10_002609	1.206776	0.083198	Xylulose kinase
EP10_002879	1.215367	0.007046	D-inositol-3-phosphate glycosyltransferase, MshA glycosyltransferase initiates biosynthesis of mycothiol (MSH), which plays a role analogous to that of glutathione
EP10_002881	1.576715	0.00023	Phosphomannomutase/phosphoglucomutase: involved in biosynthesis of lipopolysaccharide, one of the components of the bacterial outer membrane
EP10_002890	1.606423	0.000161	Phosphomethylpyrimidine synthase
EP10_003019	1.099963	0.063759	L-lactate dehydrogenase

**Table 3 ijms-25-12059-t003:** Genes whose transcription level significantly decreased (padj < 0.1) immediately after the exposure of *G. icigianus* cells to THz radiation and the response of these systems at 10 min after the end of the irradiation.

Locus ID	log2Fold Change	log2FoldChange +10 min	padj +10 min	Protein/Enzyme
EP10_000119	−3.30032	−4.03701	5.19 × 10^−16^	Copper-sensing transcriptional repressor CsoR
EP10_000120	−3.90026	−4.68006	1.25 × 10^−25^	Copper-exporting P-type ATPase
EP10_000121	−4.45124	−5.08835	1.09 × 10^−32^	Copper chaperone CopZ
EP10_000626	−1.46412	−2.53572	0.00823	Petrobactin import system permease protein YclN
EP10_000627	−1.50607	−2.64895	0.008339	Petrobactin import system permease protein YclO
EP10_000628	−1.78528	−3.07797	7.95 × 10^−5^	Petrobactin import ATP-binding protein YclP
EP10_000630	−1.96985	−2.69269	0.007614	Petrobactin-binding protein YclQ
EP10_000812	−1.93196	−3.19184	2.09 × 10^−6^	Fe^3+^ citrate-binding protein YfmC
EP10_000813	−1.66508	−2.88569	0.000111	Putative siderophore transport system permease protein YfiZ
EP10_000814	−1.70625	−3.04779	8.12 × 10^−5^	Putative siderophore transport system permease protein YfhA
EP10_000819	−1.82696	−2.79983	0.000352	High-affinity heme uptake system protein IsdE
EP10_000820	−1.70853	−2.54879	0.000931	Hemin transport system permease protein HmuU
EP10_000821	−1.66422	−2.65567	2.95 × 10^−5^	Petrobactin import ATP-binding protein FpuC
EP10_000822	−1.44776	−2.64116	2.95 × 10^−5^	Heme-degrading monooxygenase
EP10_000839	−1.5273	−2.33769	0.000601	High-affinity zinc uptake system binding-protein ZnuA
EP10_000667	−0.96761	−1.63433 *	0.182263	Putative siderophore transport system permease protein YfhA
EP10_000668	−1.01951	−1.33697 *	0.341035	Putative siderophore transport system ATP-binding protein YusV
EP10_002301	−1.04491	−1.22492 *	0.430575	Cadmium-transporting ATPase
EP10_003141	−0.9444	0.18665 *	0.924512	GTP 3′,8-cyclase
EP10_003318	−0.68941	−0.48287 *	0.516732	Beta-barrel assembly-enhancing protease

* insignificant values (padj > 0.1).

**Table 4 ijms-25-12059-t004:** Genes whose transcription level turned out to be significantly elevated (padj < 0.1) 10 min after THz irradiation of *G. icigianus* cells.

Locus ID	log2FoldChange	padj	Protein/Enzyme
EP10_000138	1.355826	0.000163	Putative amino acid permease YhdG
EP10_000693	0.863538	0.084168	5-oxoprolinase subunit A
EP10_001536	0.875392	0.066384	Glutamine-binding periplasmic protein
EP10_002112	0.894702	0.098217	Threonine synthase
EP10_002113	1.012461	0.041217	Homoserine kinase
EP10_002156	2.199392	7.13 × 10^−6^	C4-dicarboxylate transport protein
EP10_002476	0.95882	0.041329	HTH-type transcriptional regulator BetI
EP10_002477	1.006568	0.096249	Pyrrolidone-carboxylate peptidase
EP10_002622	1.222678	0.000381	Glutamine transport ATP-binding protein GlnQ
EP10_002623	1.151767	0.003803	ABC transporter glutamine-binding protein GlnH
EP10_002624	1.383302	0.00058	Putative glutamine ABC transporter permease protein GlnM,
EP10_002625	1.382731	0.001782	Putative glutamine ABC transporter permease protein GlnP

**Table 5 ijms-25-12059-t005:** Genes and operons whose levels of transcription declined 10 min after the end of exposure of *G. icigianus* cells to THz radiation.

Locus ID	log2FoldChange	padj	Protein/Enzyme
EP10_000019	−2.21125	0.000559	Chaperone protein ClpB
EP10_000089	−1.18119	0.079146	Anhydro-N-acetylmuramic acid kinase
EP10_001011	−1.9203	0.013503	Putative universal stress protein
EP10_001408	−1.54233	0.057282	Leucine dehydrogenase
EP10_001409	−1.53638	0.055198	Butyrate kinase 2
EP10_001411	−1.15153	0.054138	2-oxoisovalerate dehydrogenase. subunit alpha 2-oxoisovalerate dehydrogenase. subunit beta
EP10_001412	−1.04791	0.01829	
EP10_001436	−1.11329	0.045419	Cell division protein DivIC
EP10_001437	−1.12787	0.019737	Spore protein YabQ
EP10_001438	−1.33076	0.001686	Spore protein YabP
EP10_001951	−1.91752	0.000381	ATP-dependent Clp protease proteolytic subunit
EP10_001975	−1.13646	0.041217	Putative lipoprotein YvcA
EP10_002431	−1.88604	0.06734	Catalase-peroxidase
EP10_002805	−1.63302	0.019945	ADP-L-glycero-D-manno-heptose-6-epimerase
EP10_003512	−1.92032	2.19 × 10^−5^	HTH-type transcriptional activator Btr
EP10_003549	−1.70427	0.004183	Transcriptional regulator CtsR
EP10_003550	−1.60091	0.010916	Protein-arginine kinase activator protein
EP10_003551	−1.74359	0.001782	Protein-arginine kinase
EP10_003552	−1.70436	0.016059	Negative regulator of genetic competence ClpC/MecB
EP10_003553	−1.83009	0.003819	DNA repair protein RadA

**Table 6 ijms-25-12059-t006:** Localization of potential binding sites for metal-dependent TFs in promoters of the operons highly sensitive to THz irradiation in *G. icigianus*.

Operon	Loci	Function	log2Fold Change	Motifs of TF-Binding Sites	Location of TF Site	Position of Promoter and/or σ70-Binding Site
75	EP10_000119 EP10_000120 EP10_000121	copper export	−3.3/−4.45	CsoR Fur/PerR	−38/−24 −165/−158 −179/−171 −217/−208 −252/−234	−53/−24
319	EP10_000626 EP10_000627 EP10_000628 EP10_000629 EP10_000630	iron import	−1.46/−1.97	Fur/Zur/PerR	−295/−282 −250/−235	−53/−13
438	EP10_000812 EP10_000813 EP10_000814 EP10_000815	iron import	−1.7/−1.9	Fur/Zur/PerR	−49/−35 −80/−67 −142/−134	−86/−56
442	EP10_000819 EP10_000820 EP10_000821 EP10_000822	iron import	−1.45/−1.83	Fur/Zur/PerR	−44/−26 −81/−68 −333/−309	−89/−59
454	EP10_000839	zinc import	−1.53	Zur Fur/Zur	−61/−43 −254/−235	−70/−37

**Table 7 ijms-25-12059-t007:** Radiation parameters of Novosibirsk Free Electron Laser.

Parameters	Value, Units
Wavelength	8–400 μm
Pulse duration	50 ps
Pulse repetition frequency	2.8–11.2 MHz
Average power	up to 400 W
Peak power	up to 1 MW
Minimum relative linewidth	3 × 10^−3^

## Data Availability

The authors confirm that the data supporting the findings of this study are available within the article.

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
