# Peer review of "The Transcriptomic Response of Cells of the Thermophilic Bacterium Geobacillus icigianus to Terahertz Irradiation"

_ijms, 2024, doi:10.3390/ijms252212059_

Round 1

Reviewer 1 Report

Comments and Suggestions for Authors

Peltek et al. investigated the transcriptomic response of Geobacillus icigianus cells to terahertz (THz) irradiation, analyzing gene expression changes immediately after exposure and again 10 minutes later. The results reveal that THz irradiation affects genes involved in metal homeostasis (copper, iron, zinc) and disrupts normal cellular functions without activating heat shock genes, indicating a non-thermal response. Genes related to peptidoglycan recycling, redox reactions, and DNA/protein protection are downregulated, while systems for transition metal homeostasis show heightened sensitivity. The findings suggest that THz radiation impacts G. icigianus metabolism, likely through disruptions in metal-sensitive transcription factors. This study provides valuable insights into the non-thermal transcriptomic response of G. icigianus to terahertz radiation, with significant disruptions observed in metal homeostasis and stress response pathways. However, some areas require minor revisions:

- To deepen understanding of THz radiation effects, comparing G. icigianus transcriptomic data with other thermophiles or model organisms exposed to THz radiation could reveal unique regulatory features and adaptive response patterns in G. icigianus.

- In Line 571, it would be beneficial to specify which Illumina instrument was used for DNA sequencing to ensure methodological transparency.

- Further detail in Section 4.4 on the genome assembly and annotation process would enhance reproducibility and allow other researchers to accurately assess these methods.

- A concluding section is recommended to summarize key findings and potential applications of the study, enhancing clarity and reinforcing the study’s contributions.

Author Response

Comments 1:

Peltek et al. investigated the transcriptomic response of Geobacillus icigianus cells to terahertz (THz) irradiation, analyzing gene expression changes immediately after exposure and again 10 minutes later. The results reveal that THz irradiation affects genes involved in metal homeostasis (copper, iron, zinc) and disrupts normal cellular functions without activating heat shock genes, indicating a non-thermal response. Genes related to peptidoglycan recycling, redox reactions, and DNA/protein protection are downregulated, while systems for transition metal homeostasis show heightened sensitivity. The findings suggest that THz radiation impacts G. icigianus metabolism, likely through disruptions in metal-sensitive transcription factors. This study provides valuable insights into the non-thermal transcriptomic response of G. icigianus to terahertz radiation, with significant disruptions observed in metal homeostasis and stress response pathways. However, some areas require minor revisions:

- To deepen understanding of THz radiation effects, comparing G. icigianus transcriptomic data with other thermophiles or model organisms exposed to THz radiation could reveal unique regulatory features and adaptive response patterns in G. icigianus.

- In Line 571, it would be beneficial to specify which Illumina instrument was used for DNA sequencing to ensure methodological transparency.

- Further detail in Section 4.4 on the genome assembly and annotation process would enhance reproducibility and allow other researchers to accurately assess these methods.

-A concluding section is recommended to summarize key findings and potential applications of the study, enhancing clarity and reinforcing the study’s contributions.

Response 1:

We agree with the comments.

- To deepen understanding of THz radiation effects, comparing G. icigianus transcriptomic data with other thermophiles or model organisms exposed to THz radiation could reveal unique regulatory features and adaptive response patterns in G. icigianus.

Section 1 of this work presents data on the impact of THz radiation on the model microorganism Escherichia coli, obtained by the authors using the same source of THz radiation.

- In Line 571, it would be beneficial to specify which Illumina instrument was used for DNA sequencing to ensure methodological transparency.

Done

 In Line 587 indicated that Illumina NextSeq550 was used for DNA sequencing

- Further detail in Section 4.4 on the genome assembly and annotation process would enhance reproducibility and allow other researchers to accurately assess these methods.

Added a graphical diagram of the experiment in Section 4, which describes all stages of the work, including genome assembly and annotation

-A concluding section is recommended to summarize key findings and potential applications of the study, enhancing clarity and reinforcing the study’s contributions.

The authors believe that the “discussion” section mainly contains conclusions about the short-term and long-term response of the thermophilic bacterium genome to exposure to THz radiation. Including a detailed description of the system for controlling the homeostasis of transition metals. In this regard, introducing a separate conclusions section is impractical.

Reviewer 2 Report

Comments and Suggestions for Authors

The article is devoted to an interesting and important issue regarding the transcriptomic changes that occur in response to exposure to terahertz radiation. It turns out that there are crossover effects that suggest heating of the samples, and in fact, it is suspected that transcriptomic effects are reported that are not caused by THz radiation but by temperature shock. Therefore, the authors chose as a model organism a thermophilic microorganism, Geobacillus icigianus, whose transcriptome was analyzed immediately after THz irradiation and 10 min later.

The paper contains many experimental results that are rich in evidence.

I have the following questions and comments for the authors:

1. Although the topic of the work is terahertz radiation and its influence on living organisms, I believe that more information about G. icigianus should be given in the introduction. What attracted your attention to this organism? What could be its application as an extremophile? What are the features of its genome? Does it have unique properties? Does it have an industrial application? Please provide information on the bacterial strain you worked with.

2. Do you have a hypothesis about what the "target" of terahertz radiation in bacterial cells is? Which cell organelles or processes are attacked first or most severely? If there is no such information, mention it.

3. What prompted the re-sequencing of the Geobacillus icigianus genome and why was the originally sequenced genome not used in the comparisons? What improvements have come with the new sequencing and what missing genes have you filled in for example?

4. In the "Results" section, it is good to show statistics about the transcriptome study: how many genes changed their expression in total after the shock impact, how many are upregulated, how many are downregulated, and to what extent.

5. In Table 3 - blank fields should be noted why they are blank - no change in expression after 10 minutes? If so, mark it below the table with a sign.

6. In Table 6: based on what measurement or other indicator did you rank "Sensitivity to THz radiation" with a different number of "+"?

7. The article suffers from a lack of any illustrative material. Please make diagrams of the workflow, the irradiation setup, or a diagram of gene systems controlled by metal-sensitive transcription factors.

8. A small note: the Latin names of bacteria should be italicized in the manuscript.

Author Response

Comments 1:

The article is devoted to an interesting and important issue regarding the transcriptomic changes that occur in response to exposure to terahertz radiation. It turns out that there are crossover effects that suggest heating of the samples, and in fact, it is suspected that transcriptomic effects are reported that are not caused by THz radiation but by temperature shock. Therefore, the authors chose as a model organism a thermophilic microorganism, Geobacillus icigianus, whose transcriptome was analyzed immediately after THz irradiation and 10 min later.

The paper contains many experimental results that are rich in evidence.

I have the following questions and comments for the authors:

  1. Although the topic of the work is terahertz radiation and its influence on living organisms, I believe that more information about G. icigianus should be given in the introduction. What attracted your attention to this organism? What could be its application as an extremophile? What are the features of its genome? Does it have unique properties? Does it have an industrial application? Please provide information on the bacterial strain you worked with.
  2. Do you have a hypothesis about what the "target" of terahertz radiation in bacterial cells is? Which cell organelles or processes are attacked first or most severely? If there is no such information, mention it.
  3. What prompted the re-sequencing of the Geobacillus icigianus genome and why was the originally sequenced genome not used in the comparisons? What improvements have come with the new sequencing and what missing genes have you filled in for example?

- A concluding section is  the study’s contributions.4. In the "Results" section, it is good to show statistics about the transcriptome study: how many genes changed their expression in total after the shock impact, how many are upregulated, how many are downregulated, and to what extent.

  1. In Table 3 - blank fields should be noted why they are blank - no change in expression after 10 minutes? If so, mark it below the table with a sign.
  2. In Table 6: based on what measurement or other indicator did you rank "Sensitivity to THz radiation" with a different number of "+"?
  3. The article suffers from a lack of any illustrative material. Please make diagrams of the workflow, the irradiation setup, or a diagram of gene systems controlled by metal-sensitive transcription factors.
  4. A small note: the Latin names of bacteria should be italicized in the manuscript.

Response 1:

We agree with the comments.

  1. Although the topic of the work is terahertz radiation and its influence on living organisms, I believe that more information about G. icigianus should be given in the introduction. What attracted your attention to this organism? What could be its application as an extremophile? What are the features of its genome? Does it have unique properties? Does it have an industrial application? Please provide information on the bacterial strain you worked with.

Based on the results of preliminary experiments to study the response to THz radiation, strain G1w1 Geobacillus icigianus was selected from the collection of thermophilic bacteria from the Institute of Cytology and Genetics SB RAS as the most suitable in a number of parameters: highest growth rate, highest protein concentration, minimal sporulation.

The thermophilic gram-positive microorganism Geobacillus icigianus,strain G1w1 was isolated and described (Bryanskaya et al., 2015), and its genome was completely sequenced at the Institute of Cytology and Genetics (Bryanskaya et al., 2014). The optimal cultivation temperature for G. icigianus is 60°C. This makes it possible to separate the thermal and non-thermal components of the impact of terahertz (THz) radiation on living objects. Some features of the metabolism of this strain of G. icigianus and its possible industrial use as an oil destructor and producer of 2,3-butanediol have been studied [Tourova et al., 2016, Kulyashov et al.,2020].

  1. Do you have a hypothesis about what the "target" of terahertz radiation in bacterial cells is? Which cell organelles or processes are attacked first or most severely? If there is no such information, mention it.

Section 1 of this work presents data on the impact of THz radiation on the model microorganism Escherichia coli, obtained by the authors using the same source of THz radiation [Demidova et al., 2013, 2016, Serdyukov et al., 2020, 2021, Peltek et al., 2021.].

The proteomic response of Geobacillus icigianus was studied by the authors in Bannikova et al., 2022. In this work we showed non-thermal THz radiation affects extremophilic Geobacillus icigianus on various metabolic pathways, including disruption of cell wall synthesis, central metabolism, transcription, translation, and electron transport. We imply that DNA reparation and methylation systems are also affected by THz radiation.

  1. What prompted the re-sequencing of the Geobacillus icigianus genome and why was the originally sequenced genome not used in the comparisons? What improvements have come with the new sequencing and what missing genes have you filled in for example?

The original assembly featured less continuity. Table number 1 in the text of the article shows the statistics of the two assemblies.

Additional genes, or rather their products, found in the new assembly compared to the old one:

 2-dehydropantoate 2-reductase N-terminal domain-containing protein

 ABC-F type ribosomal protection protein

 bacteriocin immunity protein

 class 1 isoprenoid biosynthesis enzyme

 DNA endonuclease

 DNA mismatch repair protein MutT

 DUF262 domain-containing protein

 DUF3221 domain-containing protein

 DUF4288 domain-containing protein

 DUF4879 domain-containing protein

 DUF6677 family protein

 EYxxD motif small membrane protein

 flagellar basal body rod protein

 glycerophosphodiester phosphodiesterase family protein

 immunity 22 family protein

 immunity 70 family protein

 immunity protein YezG family protein

 IS481 family transposase

 lecithin retinol acyltransferase family protein

 M17 family peptidase N-terminal domain-containing protein

 MafI family immunity protein

 multidrug efflux SMR transporter

 phosphohydrolase

 RHS repeat-associated core domain-containing protein

 SagB/ThcOx family dehydrogenase

 type II restriction endonuclease

 VrrA/YqfQ family protein

 YcaO-like family protein

- A concluding section is  the study’s contributions.                                                                                4. In the "Results" section, it is good to show statistics about the transcriptome study: how many genes changed their expression in total after the shock impact, how many are upregulated, how many are downregulated, and to what extent.

Done

In Line 115 added:”To the THz irradiation, cells of the thermophilic bacterium G. icigianus responded by both increasing (after irradiation log2FoldChange>1 – 76, of which 22 are significant, at 10 min after irradiation log2FoldChange>1 - 34, of which 14 are significant) and decreasing (after irradiation log2FoldChange<-1 - 32, of which 30 are significant, at 10 min after irradiation log2FoldChange<-1 - 158, of which 56 are significant) the expression of genes”.

  1. In Table 3 - blank fields should be noted why they are blank - no change in expression after 10 minutes? If so, mark it below the table with a sign.

Done

The expression levels of these genes change insignificantly 10 minutes after irradiation

Added these values ​​to Table 3, mark it below the table with a sign

  1. In Table 6: based on what measurement or other indicator did you rank "Sensitivity to THz radiation" with a different number of "+"?

The sensitivity to terahertz of a particular operon is determined by the level (range) of changes in the expression of the genes included in it.  Taking into account your comment, we have made changes to Table 6

Replaced "Sensitivity to THz radiation" with directly “log2FoldChange”, indicating a specific range of expression level changes

  1. The article suffers from a lack of any illustrative material. Please make diagrams of the workflow, the irradiation setup, or a diagram of gene systems controlled by metal-sensitive transcription factors.

Done

 Added a graphical diagram of the experiment(Figure.2) in Section 4, which describes all stages of the work, including genome assembly and annotation to section 4 and a diagram of proposed mechanism of action of THz irradiation on Fur-dependent operons (Figure.1) to section 2.3

  1. A small note: the Latin names of bacteria should be italicized in the manuscript.

Done

All Latin names of bacteria were corrected to italics